

# A case study of rainfall-derived infiltration and inflow on a separate sanitary sewer system

Nelson J. G. Carriço[1], Rita Brito[2], Manuel Baptista[1]
[1]Barreiro School of Technology, Polytechnic Institute of Setubal, Rua Américo da Silva Marinho, 2839-001 Lavradio, Portugal
[2]National Laboratory of Civil Engineering, Avenida do Brasil, 101, 1700-066 Lisbon, Portugal
*Correspondence to*: Nelson J. G. Carriço (nelson.carrico@estbarreiro.ips.pt)
**Abstract.** Rainfall-derived infiltration and inflow (RDII) can interfere with the performance of domestic wastewater drainage
systems. It is also a major cause for the deterioration of the functional performance in those systems and for the occurrence
of domestic wastewater untreated discharges to the water environment. In most cases, the actual size and location of these
inflows are unknown. To assess this subject of RDII, a detailed knowledge of the network is required as well as a diagnosis of
the problem, namely, the type of inflows, the magnitude of their occurrence and the location of the most relevant impacts. This
paper presents the application of a methodology to estimate RDII on a Portuguese case study.
**1 Introduction**
Separate wastewater sewer systems are designed to convey sanitary and stormwater in separate sewers. There are three major
components of wastewater flow in a sanitary sewer system: base sanitary flow, groundwater infiltration and rainfall-derived
inflow and infiltration (RDII), more commonly referred to as inflow (EPA, 2014). This inflow may be considered excessive
when it compromises the systems performance or when the cost for its transport and treatment exceeds the cost to eliminate it.
Often, excessive inflow is collected during rainfall via illicit connections from roof leaders, house drains, sump pumps, or
stormwater sewers, as well as through defects in pipes and manholes (Harold, 2007). Sewers which are found connected from
the stormwater drainage system must be disconnected and rerouted as soon as possible.
After a strong rainfall event, performance of separate sanitary sewer systems may decrease significantly due to inflows which
can cause, among others: an increase of operation and capital costs of sewers and wastewater treatment plants; a decrease of
pipe capacity which potentiates untreated wastewater discharges, and consequently increases pollution; occurrence of floods
(Amorim *et al*., 2007).
Despite the significant investments made in the last decade in Portugal, in many cases wastewater systems performance is far
from satisfactory, with the perception that RDII largely contribute to this situation (Almeida e Cardoso, 2010). This problem
is well known by wastewater utility managers, who recognize that it is an important cause of functional performance
deterioration. Thus, it is essential to adopt appropriate methodological approaches and to select suitable actions to promote the
gradual reduction of RDII, in order to increase system efficiency and effectiveness in economic, environmental and operational
terms.
This paper presents the application of an estimation methodology of RDII to a real case study.
**2 Rainfall-derived inflow and infiltration, modelling and performance evaluation**
Undue connections between drainage systems are not easily located, since they are generally not registered. Monitored data is
a very valuable source of information on the systems behaviour. Undue connections can be perceived through the analysis of
flow measurement records, where the method to be used depends on whether one wants to detect connections from stormwater
to sanitary pipes or vice versa. In the first case, there are abrupt variations in the hydrograph when it rains. Mathematical
modelling can be used as a step forward to locate the undue connection, which can be followed by visual or CCTV inspections.

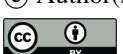



In the second case, dry weather measurements in the stormwater system detect undue permanent flows when rain events don't
occur. If the pipe is subject to water table fluctuations, these campaigns must be carried out in the dry season, where infiltration
flow is as low as possible. Once again, modelling can be used to understand how spread out infiltration is within the system.
Hydraulic modelling is an important instrument for drainage systems diagnosis (Rauch *et al.*, 2002), and can be used to analyse
existing sewer systems, to identify potential problems, and to design best corrective solutions (Nicklow *et al.*, 2004). Modelling
enables analysing how the system functions, by reproducing its actual behaviour or by estimating the values of hydraulic
variables according to pre-established scenarios, over time and along every pipe. As a support for utilities daily decisions,
models allow to locate critical pipes, support the diagnosis of system performance or enable studying alternative operational,
maintenance or rehabilitation solutions or comparing competing new projects. However, its potential can only be fully
exploited if hydraulic measurements are performed, preferably continued and obtained over a period of time that allows to
encompass different states of the system. Without calibration and validation with local data, a model is only a hypothesis for
the simulation of reality (Cardoso, 2008).
Monitored data is therefore a valuable input for modelling also, which requires hydraulic pipe data and precipitation data.
Measuring flow rates for quantifying infiltration is particularly difficult, especially when dealing with reduced flow rates or
water height. Estimating infiltration flows can also be done through the analysis of flow hydrographs or by using tracers, such
as floating solids, fluorescent liquids or chemical tracers (De Bénédittis, 2004; Kracht *et al.*, 2003). The minimum values of
the hydrographs are related to sanitary discharges at night, to the time necessary to flow upstream connections and to the
groundwater infiltration. Minimum flows can also vary over time, due to the rise of the water table after a rainy period. De
Bénédittis (2004) presents a more complete framework for the determination of infiltration and exfiltration. Cardoso (2008)
presents values related to these occurrences from bibliography and from regulatory and legislative limits present in several
countries.
Choosing the rain events to model has to address a few concerns. To start, any record above zero in the precipitation records
may be considered a rain event. However, if it has reduced intensity and duration, then surface runoff may not occur, whenever
the precipitated water volume is lower than infiltration or surface retention volumes. Even if surface runoff occurs, it may not
be totally intercepted by the drainage system under study. In this case, it is important to identify, among the available rain
events, which ones are of interest for evaluating the effects of precipitation in the drainage system (that result in changes in
the hydrograph). In WaPug (2002), Jørgensen et al. (1997) and Bertrand-Krajewski *et al.* (2000), recommendations are made
regarding which characteristics rain events should have (in intensity, total precipitation and event duration) to be identified as
relevant for modelling RDII. In addition, it is also recommended to group the precipitation events according to the impact on
the pipe flow (Saul, 1997), that is, to analyse the flow hydrograph first and then to group the rain events that have occurred
since the flow moved away from the dry weather pattern until it returns to this pattern. When there are water retentions in the
system or the infiltration component assumes a significant proportion, a flow component is registered in the hydrograph,
frequently some days after the rain event.
Performance assessment is a means for objectively quantifying the capabilities and deficiencies of the systems, supporting
decision making (Cardoso, 2008). It is supported on performance indicators, which are quantitative efficacy and efficiency
metrics that allow the diagnosis of the current situation of the infrastructure. The characteristics of the available data
determines the scale of the assessment: for a less detailed assessment, on a global scale, it is possible to use only monitoring
data; a detailed assessment, on a pipe scale, requires modelling results.
The Portuguese decree law on the design of urban water infrastructures establishes design criteria for the beginning of operation
and for the project horizon. These criteria are of a constructive nature (for example, establishing minimum and maximum pipe
slopes) and of a hydraulic nature (for example, establishing minimum and maximum velocities). Some of the legal criteria, in
particular those of a hydraulic nature, can be used to assess, at any given moment, how the infrastructure is performing.



Although constructive criteria do not in themselves constitute performance appraisal criteria, they are often the explanatory
factors for any performance shortfall in a non-performing system.
Cardoso (2007) defined twenty-six performance indicators, of which eleven relate to hydraulic performance and fifteen to
environmental performance for the evaluation of the technical performance. Estimating rainfall derived inflow and infiltration
was one of the criteria under consideration.2 Methodology
As mentioned, RDII can be assessed in different ways. In the case study, a two-step based methodology was adopted, namely
i) hydraulic modelling and assessment and ii) calculation of performance indicators.

*2.1 Hydraulic modelling and assessment*

There are several computer applications available in the market, commercial or freeware, that allow building hydraulic models
for drainage systems
In the case study, Storm Water Management Model (SWMM) from Environmental Protection Agency (EPA) was used.
SWMM is used throughout the world for planning, analysis and design related to stormwater runoff, combined and sanitary
sewers, and other drainage systems in urban areas. Furthermore, SWMM is freeware and has a vast user community that
participates intensely in forums and newsgroups.
The use of SWMM implies collecting several data such as sewers and manholes registry information, amount of sewage
discharged per manhole and flow measurements. The discharge per manhole can be estimated from the product of population,
per capita water use and ratio of water rejected to the sewer system. Flow measurement can be used to calibrate the hydraulic
model. Furthermore, it is necessary to collect precipitation data to investigate RDII in a separate sanitary sewer system.
The hydraulic model can be used for dry-weather flow and for contributing impervious area determination. The former is
generally used to establish a dry-weather flow pattern from flowmeter data. The latter is used to estimate the amount of
impervious area wrongly connected to the sanitary sewer pipes (Brito et al., 2009). When facing a separate sewer system with
unwanted stormwater sewer connections, the contributing impervious area is difficult to quantify. Being so, in such systems it
may not be useful to accurately determine the real impervious areas. This would mean that the whole amount of net
precipitation would drain to the pipes, which is of course not correct in the case of sanitary sewer systems (Brito *et al*., 2009).
The part of runoff that derives from the hydrological model could be determined based on the rainfall volume (R-value)
method, which calculates RDII volume as a fixed percentage of the rainfall amount. Based on the R-value method and the
Rational Method, it is feasible to adopt an auxiliary parameter, contributing impervious area A', suitable to estimate extraneous
rain water flows. A' represents the amount of impervious area connected, a parameter that varies with the rainfall event (Brito
*et al*., 2009).
For a proper characterization of pipe behaviour, it is considered that at least two days of dry weather flow and three precipitation
events should be modelled, and this number should be higher if dry weather patterns show significant variations or if undue
inflows of pluvial origin occur in sanitary systems (either direct inflows or infiltration) (WaPug, 2002).

*2.2 Calculation of performance indicators*

As referred, performance Indicators (PIs) provide key information to assess the efficiency and effectiveness of a service, and
may thus be used as a measure of a particular aspect of an utility's performance or standard of service (Matos *et al*., 2003). PIs
are considered to be a means of aggregating information on system characteristics and data gathered from monitoring or
modelling and translated into performance values (Cardoso and Frehmann, 2010). PIs can be classified in relation to "good"
or "bad" performance.



Table 1 shows the Performance Indicators for RDII used in the case study (from Cardoso, 2008).





**Table 1 - Performance Indicators for RDII**

| ID | Designation | Definition | Unit | Performance | | |
|---|---|---|---|---|---|---|
| | | | | Good | Average | Bad |
| $PI_1$ | Proportion of the sewer full section flow capacity ($Q_{full}$) used by the minimum daily dry-weather flow ($Qmin_{dw}$) | $\dfrac{Qmin_{dw}}{Q_{full}}$ | % | <25 | 25-50 | >50 |
| $PI_2$ | Proportion of the sewer full section flow capacity ($Q_{full}$) used by the maximum daily dry-weather flow ($Qmax_{dw}$) | $\dfrac{Qmax_{dw}}{Q_{full}}$ | % | <75 | 75-100 | >100 |
| $PI_3$ | Proportion of minimum daily dry-weather flow ($Qmin_{dw}$) by average daily dry-weather flow ($Qavg_{dw}$) | $\dfrac{Qavg_{dw}}{Qmin_{dw}}$ | % | <25 | 25-50 | >50 |
| $PI_4$ | Ratio between maximum daily dry-weather flow ($Qmax_{dw}$) and average daily dry-weather flow ($Qavg_{dw}$) | $\dfrac{Qmax_{dw}}{Qavg_{dw}}$ | - | <3 | 3-5 | >5 |
| $PI_5$ | Minimum daily dry-weather flow ($Qmin_{dw}$) per unit length of sewer ($Lsewer$) | $\dfrac{Qmin_{dw}}{Lsewer}$ | m³/(day.km) | <40 | 40-80 | >80 |
| $PI_6$ | Proportion of the sewer full section flow capacity ($Q_{full}$) used by the maximum daily wet-weather flow ($Qmax_{ww}$) | $\dfrac{Qmin_{ww}}{Q_{full}}$ | % | <75 | 75-100 | >100 |
| $PI_7$ | Proportion of daily wet-weather volume ($V_{ww}$) by daily dry-weather volume ($V_{dw}$) | $\dfrac{V_{ww}}{V_{dw}}$ | - | <3 | 3-6 | >6 |

## 120 3 Case study

### 121 3.1 System description

The case study is a small sanitary sewer system located in Lisbon Metropolitan Area, Portugal. The urban catchment has a
very heterogeneous occupation with buildings up to 5 floors, a supermarket, a police station, schools and some shops.
Estimated residential population is about 4,018 inhabitants. The sewer network is about 11.8 km long and has around 1,577
domestic and 205 non-domestic service connections. Most sewers are made of PVC with 200 mm of diameter (which is
minimum diameter allowed by the Portuguese design decree law). The system has a pumping station that raises wastewater
from a lower to a higher elevation of the network. At the final end of the system there is a flowmeter that measures the amount
of wastewater delivered by the Municipality to a public company that will convey it to the wastewater treatment plant. A study
made by Municipality concluded that the total amount of potable water sold in the urban catchment of the case study was lower
in about 56% than the wastewater delivered to the company for treatment. This is a huge difference that can only be explained
by some kind of RDII.
From a hydrogeological point of view, Portugal is a favoured country and major groundwater unit of the Iberian Peninsula is
the huge Tagus-Sado aquifer system. Despite the case study is located above the Tagus-Sado aquifer system in a first approach,
groundwater infiltration was not utility's major concern and the study will focus mainly on the inflow component.

### 135 3.2 Flow and precipitation data

The flowmeter installed downstream the system measures the amount of wastewater delivered by the Municipality to the
wastewater treatment plant (WWTP). Flow data was available for a period of 181 days (from 5th June 2014 to 2nd December
2014), and has a 15 minutes time step. . This time step is not the most suitable to allow a comparative analysis with precipitation
data, once precipitation events have high variability within 15 minute periods.
Precipitation data from a rain gauge temporarily installed in a neighbouring municipality was available. Despite the rain gauge
not being located in the catchment area, it was the only in the surroundings where data with a time step smaller than 1 hour
was accesible. Precipitation data for a period of 1 year (from 3rd February 2014 to 4th February 2015) was gathered.



The annual climatologic bulletins of 2014 and 2015 from the Portuguese atmospheric and ocean institute evidence that 2014
was the rainiest of the last 25 years and 2015 was the sixth dryer year since 1931 and the fourth since 2000, respectively.
Since flow data refers only to the second half of 2014, the analysis will merely focus on this period.
**3.3 Hydraulic modelling and assessment**
The hydraulic model was built based on information provided by the Municipality which included registry data from the pipes
and manholes. Some additional verifications were necessary, since some data were inexistent (e.g. pipes without diameter) or
were wrongly attributed (e.g. pipes with negative slope). A total of 443 manholes and corresponding pipes were modelled.
The Portuguese design decree law has several dimension criteria to be checked, such as maximum length between manholes
($L_{max}$ = 60 m), minimum sewer diameter ($DN_{min}$ = 200 mm), minimum and maximum slope ($i_{min}$ and $i_{max}$, 0,5 and 15%,
respectively) and minimum depth ($h_{min}$= 1,2 m). Table 2 shows the percentage of pipes that don't comply with design criteria.
**Table 2 – Percentage of sewers in noncompliance with dimension design criteria**

| Design criteria | $L_{max}$ | $DN_{min}$ | $i_{min}$ | $i_{max}$ | $h_{min}$ |
|---|---|---|---|---|---|
| Noncompliant sewers (%) | 0.7 | 0.0 | 19.4 | 1.1 | 3.0 |


Most criteria are complied with. Nevertheless, the minimum slope is a concern in this system, which may limit drainage
capacity. For each of the criteria, noncompliant pipes location was identified in a report for the utility.
The amount of sewage discharged per manhole is estimated considering population, per capita drinking water use and ratio of
water rejected to the sewer system. According to the Portuguese 2011 Census, resident population in the urban catchment was
4,018 inhabitants. To estimate the average sanitary flow, drinking water consumptions in 2008 and 2012 were used. Registered
values of 193,260.5 m³ and 160,688.6 m³ correspond to a per capita water use of 132 and 110 L/(inhabitant.day), respectively
for 2008 and 2012. The Portuguese design decree law refers, for design purposes, that circa 80% of drinking water generates
sanitary wastewater; therefore, the corresponding per capita sewage was 105 L/(inhabitant.day) and 88 L/(inhabitant.day) for
2008 and 2012. An average value of 100 L/(inhabitant.day) was firstly considered for modelling.
It's interesting to compare this value to the ones obtained based on the available wastewater measurements. Dividing the
wastewater volume measured in 2014 (in 181 days, 143,647.4 m³ were registered) by the resident population, a per capita
sewage of 197 L/(inhabitant.day) is obtained. This value is much higher than the others previously estimate, which may be an
indicator of the presence of RDII.
The dry weather inflows were obtain considering the contribution from each house hold, adjusted on an hourly basis by
applying the time pattern multipliers shown in Figure 1.

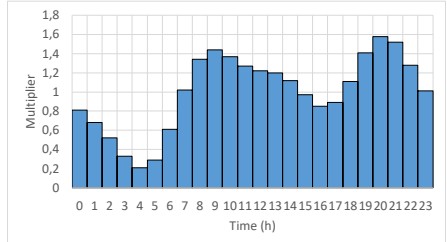


**Figure 1- Hourly dry-weather inflows pattern**
The pumping station was installed circa 20 years ago, and has two pumps, one for regular service and the other for emergency
purposes. These pumps, installed in a wet well, are vertical axis centrifugal pumps and were designed for a flow rate of 20.3
m³/hr. This pumping station was modelled in SWMM with a storage tank connected to a pump. The pump curve type chosen





was type 4, where flow varies continuously with the inlet node water depth. Two control rules were implemented, the first
starts pumping when water depth is higher than 0.5 m and the second stops pumping when water depth is lower than 0.1 m.
For dry weather, hydraulic design criteria were evaluated, such as minimum and maximum velocity ($v_{min}$ and $v_{max}$, 0,6 and 3
m/s, respectively) and maximum relative water depth (h/$D_{máx}$= 0,5 m). Table 3 shows the percentage of pipes that don't comply
with design criteria.
**Table 3 – Percentage of sewers in noncompliance with hydraulic design criteria**

| Design criteria | $v_{min}$ | $v_{max}$ | h/$D_{máx}$ |
|---|---|---|---|
| Noncompliant sewers (%) | 91,9 | 0 | 1,8 |


The minimum velocity is a concern in this system, which is associable to low slopes and reduced flows in the upstream pipes.
Model validation for dry-weather flow was performed through the adjustment of per capita sewage inflow and for the pump
curve parameters. The adequacy of the simulated flow in face of the measured flow was tested for the 5 days under analysis.
The adequacy was evaluated by calculating the volumetric error and by graphical comparison between the flow series. The
volumetric errors after validation vary between -7% and 4%, which was considered acceptable. For dry weather, the ratio
between the simulated and measured flow values should have a volumetric error between -10% and 10% (WaPUG, Watewater
Users Group, Allit, 1999).
After model validation for the dry-weather scenario, system behaviour during rainfall events was assessed. Drainage
subcatchments (five, from SC1 to SC5) were characterized and connected to downstream manholes. SWMM requires the
definition of the subcatchment impervious area, and assumes that all this area contributes to pipe flow. Based on a preliminary
evaluation of runoff volume during wet weather, for the case study, three different contributing impervious areas were
considered ($A_1$'=16%, $A_2$'=27% and $A_3$'=38%). For each of these cases, three rainy days were studied (R1 in 10 of September
of 2014, R2 and R3 in12 and 13 of October of 2014), resulting in nine different wet weather scenarios. For these scenarios, a
few pipes surcharge and between 4 and 48 manholes are subject to sewage discharge through the manhole cover.
**3.4 Performance indicators**
Some of the results obtained by PIs calculation are shown in Figure 2.

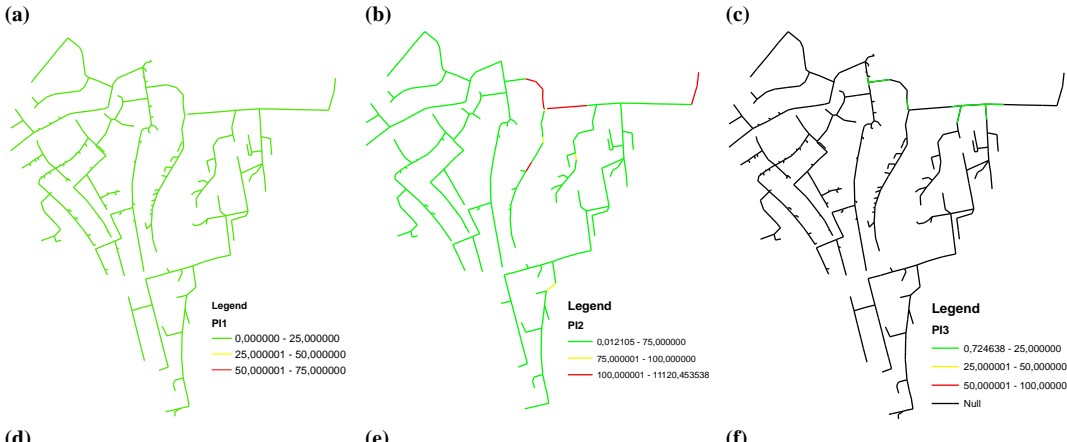



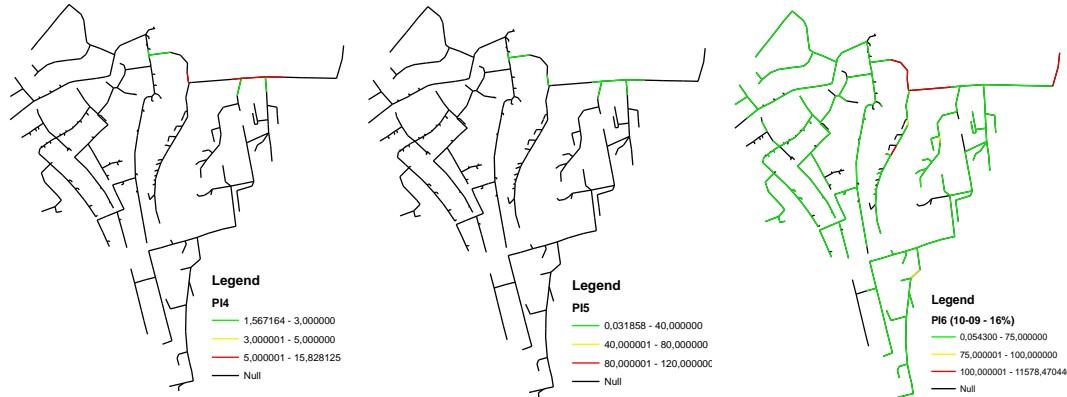

**Figure 2 - Performance Indicators for RDII: (a) PI₁; (b) PI₂; (c) PI₃; (d) PI₄; (e) PI₅; (f) PI₆ for R1 and A₁'=16% scenario**
Both PI1 and PI5 relate to minimum flow. Figure 2 (a) shows that all sewers in the network are below 25% which means a
good performance in $PI_1$. In $PI_5$, the results obtained for the upstream sections of subcatchments give values below 40
$m^3$/(day.km) which means a good performance. The lowest value obtained in PI5 was 0.31 $m^3$/(day.km) and the highest was
1.31 $m^3$/(day.km). These results indicate a low concern with infiltration.
In the case of PI2, nine sewers have a value exceeding 100% which means bad performance in this indicator. These pipes
surcharge in dry weather without any exceptional input. This behaviour may compromise system performance. Only two of
the nine sewers are within the regulatory slope and three of them have very constrained sections. Most of the sewers (i.e. 295)
present a good performance and 4 present an average performance in indicator $PI_2$ (see Figure 2b). Nevertheless, it is important
to underline that PI2 is classified as "good" for values lower than 75%, whereas the decree law stipulates as a design criterion
a maximum h/D of 50%. That is, the situation could be more critical if the IP2 reference values were more similar to those in
the decree law. With circa 1.8% of the network non complying in relation to h/D (Table 3), it is foreseeable that around 75
pipes could present limitations on IP2 if reference values were adjusted accordingly.
The indicator $PI_3$ only was computed for the sewers immediately downstream of the subcatchments. As shown in Figure 2 (c)
all sewers present an indicator below 25% which represents an overall good performance. It should be noted that four pipes
have IP3 above 20%, which, even more than IP1, may be indicative of the presence of infiltration.
$PI_4$ represents the ratio between maximum daily dry-weather flow and average daily dry-weather flow and can be associated
to a peak factor. The Portuguese design decree law establishes that for sanitary systems the peak factor is the ratio between
maximum flow and the annual average flow and can be determined by Eq. (1):
$$fp = 1,5 + \frac{60}{pop^{0,5}}$$                   (1)
in which fp = peak factor and pop = number of served inhabitants by the sanitary system.
Table 4 shows the results obtained for the five subcatchments considered in the case study,

220                   **Table 4 – Results of fp and PI₄ obtained**

| Subcatchment | Pop (inhabitants) | fp (-) | PI₄ (-) |
|---|---|---|---|
| SC1 | 378 | 4,6 | 1,58 |
| SC2 | 402 | 4,5 | 1,58 |
| SC3 | 234 | 5,4 | 15,54 |
| SC4 | 456 | 4,3 | 12,04 |
| SC5 | 2208 | 2,8 | 5,07 |

It appears that there is no direct relation between these two parameters neither in terms of magnitude nor the proportion of PI₄
with the contributing population. This observation suggests the existence of other uses in addition to the domestic uses. Like

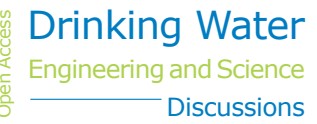

PI₃, this indicator was computed for the sewers at the downstream end of the subcatchments. Figure 2 (d) shows that two
sewers present good performance and the other three have bad performance. In these latter cases, as those don't correspond to
upstream pipes, it may be relevant to carry out a more detailed survey of the wastewater contributions, in order to acknowledge
possible water uses not known to the utility.
PI6 is computed for all sewers, except for those where flow is null (see Figure 2f), for the 9 wet weather scenarios under study.
For these scenarios, between 84% and 96% of the pipes presented a good performance, but between 3% and 14% presented
bad performance. Most pipes have good performance in PI6. Nevertheless, a bad performance in this PI means that the pipes
surcharged due to precipitation, which occurred in more than 10% of the pipes in two scenarios, which already conditions the
performance of the system.
PI₇ represents the proportion of daily wet-weather volume by daily dry-weather volume and is computed for the sewers
downstream the subcatchments, for the 9 scenarios. Results obtained show that all sewers present a good performance since
all values obtained were below 3. This means in overview, that there is no generic problem of excess of stormwater volume
from the contributing subcatchments. Nevertheless, it should not be overlooked that in every scenario there is a volume higher
than dry weather volume (PI7 ranged from 1.0 and 2.6), which means that there is an effective rainfall contribution in all
events, even if the ranks in ID7 is not concerning. What this also means is that there is a reduced proportion between the
rainfall volumes and the sanitary volumes; in the case of discharging through the manhole cover, this discharge has a dilution
of less than 1:3, which may constitute a problem from the point of view of environmental impact and public health.
In summary, considering the study of the various performance indicators, it was verified that the pipes evaluated with bad
performance mostly belong to the same area in the system, namely the northern zone where the pumping station is installed
and where two subcatchments connect. The remaining areas of the system presented acceptable performance results.
**4 Final remarks**
In this paper an application of an estimation methodology of RDII to a Portuguese case study was presented. A two-step based
methodology based in hydraulic modelling and assessment and performance evaluation was explained and applied. The aim
of the study was to estimate the amount of RDII in the sanitary system and locate the priority intervention areas mostly
concerning direct rainwater inflow. This case study is a small sanitary sewer system located in Lisbon Metropolitan Area,
Portugal. The obtained results showed that there is no major overall problem of RDII in the case study. Note that the study has
some limitations that can influence the final results such as i) the flow data collected showed values with large fluctuations in
many days' due pump operation which was difficult to model in SWMM; ii) precipitation data were available only for 2014;
iii) precipitation data were available with a time period of 15 minutes not addressing precipitation variability; and iv) the rain
gauge was not located in the system subcatchments. In future works is recommended to install a rain gauge in the system
subcatchments and collect precipitation data with a time period of 1 minute during an entire year. Additionally, infiltration
studies could be more detailed; hydrogeological studies should be carried out in order to analyse the influence of the water
table to the wastewater drainage system.

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

Group.


