# Peer review of "A case study of rainfall-derived infiltration and inflow on a separate sanitary sewer system"

_Drinking Water Engineering and Science, 2017_

## Referee Comment (RC1) · Anonymous Referee #1 · 29 Mar 2017

The paper deals with a model based analysis of a small wastewater (sewer) system in Lisbon, combined with a set of incdictors.

There are several weaknesses:

1. The selection of the system to study was not justified and it appeared that the results did not comply with the ambitions of the paper, due to system limitations and lack of data. This may be a common situation, but is not discussed.

2. The selection of indicators is not justified, it leans to one single research only

3. There are no general discussion of the results, except a few sentences in the conclusion

The paper appears as a preliminary report from a consultancy. The scientific dimension is not clear and neither discussed or justified. Some references in the opening chapter seems a little randomly picked, without a substandtal discussion of the relevance within the context.

---

## Referee Comment (RC2) · Anonymous Referee #2 · 18 Apr 2017

The paper by Carrico et al. presents the application of a standard methodology for assessing the rainfall-derived infiltration and inflow (RDII) using a case study in Portugal. Although the manuscript gathers some information on the analysis of the RDII, the literature review is very limited, and the terminology used is very often confusing (e.g. I am not sure what "undue connections" term refers to). However, my main comment and concern is the novelty of the presented work. I do not see any contribution to knowledge in the paper, the authors do not give any information on why the study is unique, what are the novel aspects of the methodology presented, nor what are the main novel insights gained from the study. As such, unfortunately I find that the manuscript is not suitable for publication in a scientific journal.

---

## Referee Comment (RC3) · Anonymous Referee #3 · 8 May 2017

The paper aims at describing the application of a two-step methodology -comprising the use of performance indicators and sewer modelling- to the analysis of rainfall-derived infiltration and inflow (RDII) into small a wastewater sewer network in Lisbon, Portugal.

The topic of the paper is interesting and indeed corresponds to a recurrent problem in sewer system operation. Furthermore, the adopted case study is also potentially very interesting. However, the paper has multiple shortcomings, with the main ones being that (1) the main aim of the study is not satisfactorily fulfilled (i.e. RDI is not effectively analysed); (2) I see no relevant contribution to knowledge in this paper. Further short-comings/details are summarised below. For these reasons, I would recommend that

the paper is not considered further for publication in a scientific journal.

Detailed comments:

Chapter 1: The introduction is very generic and does not indicate how the work presented in the paper fits within current research and why it constitutes a valuable contribution to this area.

Chapter 2: The literature review is also rather generic and often irrelevant (e.g. no need to explain what sewer modelling software). A detailed review is missing of the core processes, recent developments and on-going research in the area of sewer infiltration modelling and detection (e.g. infiltration models should be at the centre of the paper and only the (selected) rational method is vaguely mentioned, no reference is made to recent studies which aim at explicitly modelling infiltration into sewers). As a result, the selected methodologies are poorly justified.

Chapter 3:

- The description of the sewer system and available monitoring data is very poor. A map would help illustrate it. Temporal resolution of rainfall data must be specified (the authors simply mention 'smaller than 1 h' – bear in mind that for sewer modelling 15 min may already be too coarse, so 'smaller (finer) than 1 h' is not necessarily sufficient – it is only in the 'additional remarks' section where they mention the temporal resolution of rainfall inputs).

- No proper justification is given as to why ground water infiltration is not a concern and is therefore disregarded in this study.

- Most of the sewer design criteria listed in Tables 2 and 3 are irrelevant for the present study and their relevance is not discussed in the paper. It really looks as if this paper were just an excerpt of a technical report.

- Line 190: what do SC1 and SC5 mean?

- Section 3.3. does not discuss RDII at any point. It discusses other modelling elements which are mostly irrelevant for the present study, but ultimately the sewer model is not truly used for analysing RDII.

- Analysis of the relevance of performance indicators in relation to infiltration is rather limited. Many of the indicators and the way in which they are described are largely irrelevant for the purpose of the paper.

- The final conclusion in Chapter 3 is that the majority of the sewer system (except of the northern section) displays acceptable performance, but no clear reference is made to infiltration.
* * *

---

## Author Comment (AC1) · 17 May 2017

Dear referee,

we agree with your comments and we will take them into consideration if there exists a further review of the paper.

Kind regards, Nelson Carriço
* * *

---

## Author Comment (AC2) · 17 May 2017

Dear referee,

we agree that the paper is maybe not well structured and therefore your observation is accepted by us. The term undue connections is wrong, the right term should be illicit connections.We will rewrite a new version of the paper including your comments.

Best Regards, Nelson Carriço

———————————————

---

## Author Comment (AC3) · 17 May 2017

Dear referee,

we agree with your comments and we will take them into consideration if there exists a further review of the paper.

Kind regards, Nelson Carriço
* * *